# Enhancing Physical Activity and Psychological Well-Being in College Students during COVID-19 through WeActive and WeMindful Interventions

**DOI:** 10.3390/ijerph19074144

**Published:** 2022-03-31

**Authors:** Kathryn Friedman, Michele W. Marenus, Andy Murray, Ana Cahuas, Haley Ottensoser, Julia Sanowski, Weiyun Chen

**Affiliations:** 1School of Kinesiology, University of Michigan, Ann Arbor, MI 48109, USA; ksfried@umich.edu (K.F.); mmarenus@umich.edu (M.W.M.); migwetch@umich.edu (A.M.); haleyott@umich.edu (H.O.); jcsano@umich.edu (J.S.); 2Department of Psychology, University of Michigan, Ann Arbor, MI 48109, USA; acahuas@umich.edu

**Keywords:** physical activity, resilience, intensity, university students, COVID-19, web-based intervention

## Abstract

This study aimed to examine the immediate and short-term effects of aerobic and resistance training (WeActive) and mindful exercise (WeMindful) virtual interventions in improving physical activity (PA) and resilience among college students. Participants were 55 students who were randomly assigned to either the WeActive group (*n* = 31) or the WeMindful group (*n* = 24). Both groups attended two virtual 30 min aerobic and resistance training sessions (WeActive) or mindful exercise sessions (WeMindful) per week for eight weeks. All participants completed the International Physical Activity Questionnaire and the Connor–Davidson Resilience Scale (CD-RISC-10) via Qualtrics one week prior to (pre-test) and after the intervention (post-test) and 6 weeks after the intervention (follow up). There was a significant main effect of time for resilience (*F* = 3.4.15, *p* = 0.024), where both the WeActive group and the WeMindful group significantly increased the resilience scores from pre-test to follow up (*t* = −2.74, *p* = 0.02; *t* = −2.54, *p* = 0.04), respectively. For moderate physical activity (MPA), there was a significant interaction effect of time with group (*F* = 4.81, *p* = 0.01, *η*^2^ = 0.038), where the WeActive group significantly increased MPA over time from pre-test to follow-up test as compared to the WeMindful group (*t* = −2.6, *p* = 0.033). Only the WeActive intervention was effective in increasing MPA. Both interventions were effective in increasing resilience from pre-test to 6 week follow up.

## 1. Introduction

The mental health of university students has been a prominent area of study. According to a worldwide study of college students [1], one-fifth of college students had a clinically significant mental health disorder. Similarly, a study of over 2000 university students in Australia observed that one in four students reported a mental health disorder [2]. The COVID-19 pandemic has exacerbated the existing negative impacts of mental health problems due to increased anxiety about health, decreased social interaction, and increased worry about academic performance [3]. This has brought an urgent need for positive interventions to improve mental health in university students.

Resilience, defined as the ability to adapt positively after experiencing adversity, provides protection for negative mental health during challenging times [4]. Resilient people tend to be optimistic, emotionally aware, social, and able to confront issues while maintaining perspective and daily function [4]. In other words, resilience represents a strength to overcome obstacles with competence and hope. In a study of 7800 college students, it was found that resilience played a mediating role between the association of COVID-19 stressful experiences and acute stress disorder [5].

Physical activity is crucial in developing and strengthening resilience [6]. A study found a positive relationship between physical activity and resilience in college students [7]. Furthermore, a study found that PA level was significantly correlated with resilience in Chinese 7th and 8th graders [8]. Helping students engage in PA and giving resources to promote the continuation of PA engagement can thus be inferred to positively affect resilience. Despite the beneficial effect of PA on resilience, university students tend not to be engaging in the recommended amounts of exercise [9]. In a study of 296 university students, it was found that only 5.4% of students met the WHO’s PA recommendation of 150 min/week of moderate to vigorous physical activity (MVPA) or 75 min/week of vigorous PA [10]. However, virtual PA interventions may help to increase this low engagement, especially during the pandemic. In a web-based randomized control trial of mothers, it was found that after engaging in an 8 week video-based exercise intervention (5–30 min of video-based exercise 5 days per week), the intervention group increased their MVPA by 42.2 min compared to the waitlist control group [11]. In a randomized control trial of 64 young adults during the COVID-19 pandemic, an intervention group received weekly YouTube video-based aerobic and resistance exercises while a control group received general health education videos for 12 weeks [12]. The intervention group significantly increased MVPA, but did not significantly increase light PA [12].

Different intensities of PA can also impact resilience. A study of 244 undergraduate students found that the higher scores of resilience were significantly associated with increased vigorous PA [13]. Additionally, in a randomized controlled trial of 67 adults, the results indicated that both high-intensity interval training and moderate-intensity training increased resilience [14]. Similarly, study of Spanish adults early in the pandemic found that individuals who engaged in VPA during the first week of isolation reported higher resilience [15].

Mindfulness is the heightened awareness that develops through intentional non-judgmental attention to the present [16]. Mindful exercise is mindfulness with the addition of physical exertion and extra attention to proprioception [17]. Mindfulness and mindful exercise can also be a tool to build and maintain resilience. In a cross-sectional study of 106 nursing undergraduates, a strong positive relationship between resilience and mindfulness was found [18]. Moreover, mindful exercise such as yoga can be beneficial in increasing resilience. In a randomized controlled trial of 121 adolescents, it was found that a yoga intervention including simple yoga poses, breathing exercises, and visualization insignificantly increased resilience from baseline, while the control group significantly decreased their resilience [19]. Furthermore, an intervention study of 123 rural-urban migrant college students found that students who engaged in a 40 session yoga intervention reported a significantly higher total resilience score as compared to the control group [20].

Due to COVID-19 precautions, virtual interventions are at the forefront of research. A systematic review of web-based PA interventions found that most significantly increase PA in its participants. [21]. Additionally, virtual platforms provide increased flexibility and lower costs [21]. Moreover, in a systematic review of university students early in the pandemic, significant reductions in all intensities of physical activity were found [22]. As universities enact strict social distancing policies to mitigate spread of COVID-19, college students need COVID-19-safe physical activity and mindful opportunities. Virtual aerobic exercise may provide this opportunity for PA. In addition, mindfulness-based yoga may combine the benefits of physical activity and mindfulness, benefitting students physically and mentally. However, the effects of a web-based exercise and mindful intervention on resilience and all PA intensities are not well known. Therefore, the purpose of this study was to examine the immediate and short-term effects of virtual aerobic and mindful exercise interventions on physical activity intensity engagement and resilience among college students. It is hypothesized that the interventions may increase various intensity levels of PA and resilience from pre-test to post-test, and six-week follow up. Regarding intensity, it is hypothesized that the aerobic and resistance exercise group, WeActive, may increase MVPA and the mindful yoga exercise group, WeMindful, may increase walking PA from pre-test to post-test, and six week follow up. This study is significant in that the results may help inform university administrators as they plan exercise and mindfulness sessions that strive to increase the physical activity and resilience of their students.

## 2. Materials and Methods

### 2.1. Research Participants and Setting

The participants in this study were 77 students enrolled in a large midwestern university. Seventy-one students were analyzed at post-test. Three students were excluded due to lack of participation and five for missing outcome data. Among the 71, 55 students were analyzed at follow up. Fourteen students did not complete the follow up survey and two were excluded due to lack of data in the outcome measures screened using listwise deletion. Approximately 58% were undergraduate students. In total, 46 of the participants were cisgender female, 5 cisgender male and 4 transgender and gender non-conforming (TGNC). Demographic data are illustrated in Table 1. We recruited the participants by using several advertisement approaches, including posts on our lab Instagram account, the bulletin from the School of Kinesiology, and the Canvas dashboard (the learning management system), in addition to a targeted email system. Inclusion criteria for eligible participants in this study included being enrolled at the University of Michigan, giving written consent prior to participation, and possessing the physical and mental abilities to engage in the intervention and questionnaire (see Appendix A). Further, the participants were all required to have access to a device with internet access throughout the duration of the intervention and assessment periods. Exclusion criteria included being under the age of 18, and not able to exercise due to injury or illness. This study was approved by the University Institutional Review Board of Health and Behavioral Sciences (IRB#HUM00189120/Ame00107415).

A quasi-experimental design was used to assign the participants to one of the two intervention groups: WeActive or WeMindful. The intervention was implemented during Spring 2021 and lasted eight weeks. Participants completed the physical activity and resilience questionnaire at pre-test (one week prior to intervention), post-test (one week after conclusion of intervention), and follow up (six weeks after the intervention).

### 2.2. Intervention Conditions

Both WeMindful and WeActive groups attended two, 30 min exercise sessions per week over the course of eight weeks. During each week, one 30 min exercise session was held synchronously using Zoom on Monday for the WeActive group and on Tuesday for the WeMindful group. The corresponding investigator oversaw and supervised the implementation of each session via Zoom. The participants in each group were instructed to repeat the session asynchronously on Wednesday for the WeActive group and on Thursday for the WeMindful group via a recording posted on Canvas. Additionally, lesson plans were posted on Canvas for participants who rather follow along than watch a recorded video. Students were encouraged to complete the asynchronous session with peers safely if applicable. Each week, an announcement was sent through Canvas reminding participants of the exercise session times and providing positive reinforcement such as “Keep up the good work” and “Don’t forget why you started”.

#### 2.2.1. WeActive

WeActive lessons focused on aerobic and strength training exercises. When teaching each lesson, the student-instructor used the structured five–twenty–five-minute teaching format. Five minutes at the beginning and end were dedicated to warm up and cooldown, respectively. During the warmup, participants engaged in different walking patterns in the first four weeks and then high-impact aerobic movements in the last four weeks and dynamic stretches. The cool down focused on static stretches and ended with a summary review of the lesson. During the main twenty minutes of the session, the participants engaged in strength exercises through a circuit format. The strength training was focused on a mix of both muscular strength and endurance. The instructor had two circuits for each lesson. Examples of circuit focuses were cardio, legs, core, arms, standing, and floor. Moreover, examples of movements in the circuit training include wall sits, butt kicks, flutter kicks, jump squats, dead bugs, burpees, and Russian twists. The instructor started with easier exercises earlier in the intervention and slightly progressed as intervention continued. The circuits started 40 s on followed by 20 s of rest for each movement and the circuit was repeated 2–3 times. At halfway through the intervention, this was increased to 45 s on and 15 s of rest. Moreover, both circuits together included 6–10 exercises. The student-instructor first verbally explained and physically demonstrated the exercise, including modifications. Then, participants were led to perform the exercise or accommodated movement. The instructor provided cues on proper forms of the exercise and motivation throughout the lesson. The WeActive instructor was a graduate student in the movement science program at the University of Michigan. The instructor is a Certified Strength and Conditioning Specialist through the National Strength and Conditioning Association and has taught training and group exercise classes for five years prior. Further, the instructor had taught virtual sessions in the most recent year.

#### 2.2.2. WeMindful

WeMindful lessons were mixed types of mindfulness-based yoga exercises. Physical exertion, intentional attention and proprioception were emphasized. The student-instructor also used the structured lesson format for teaching each lesson. For example, each lesson started with a five-minute mindful warm up with short-term goal setting. An example of warm up activity is a body scan, where participants complete isolated movements of body parts starting at their toes and moving superiorly. Cues to become consciously aware of tactile and proprioceptive input were implemented. When the warmup movements ended, participants were instructed to take deep breaths and set a goal for the session to come. The main session of the lesson consisted of 4–6 new beginner yoga poses and a review of the past week’s poses. Examples of poses included downward dog, low lunge, tree, cat-cow, and triangle pose. When teaching most poses, the instructors provided recommended accommodations to increase accessibility. After learning each pose, the facilitator taught the participants to put the poses together into a flow. Throughout the flow, the instructor cued participants through cadenced breath, i.e., breathing in during the pose and out during the transition. Additionally, cues included engaging in the moment, being aware of the body, and breath. At the conclusion of the session, a five-minute mindfulness script was presented. The script often focused on breath, winding down and positive reinforcement. During weeks 1–3 and 5–7, participants learned new poses and flows. These poses and flows were then reviewed at the midterm session (week 4) and final session (week 8). The instructor for WeMindful lessons was an upperclassman in the movement science department at the University of Michigan. The student had two years of yoga experience and three years of experience teaching group exercise sessions such as dance and running. Since the instructor had not taught virtually prior to the intervention, they worked under the mentorship of the WeActive instructor.

#### 2.2.3. Peer Coaching

A doctoral student who has been trained in motivational interviewing and an undergraduate completing a psychology degree planned and facilitated a supplemental peer coaching session via Zoom for both the WeActive and WeMindful groups. The session volume included one session every two weeks for 30 min. The objective of peer coaching was to aid participants in goal setting, facilitate virtual social interactions, and encourage reflection of progress. In Zoom sessions, participants spoke out loud and discussed obstacles and coping mechanisms, perspectives of the exercise sessions and suggestions for future lessons. In addition to the Zoom sessions, participants engaged with journal prompts to reflect independently on their progress, goals, and general feelings throughout the intervention. All materials were posted on the intervention study website.

### 2.3. Outcome Measures

The questionnaire was administered through Qualtrics one week prior to the intervention (pre-test), one week after the intervention (post-test), and six weeks after the intervention (follow up). The questionnaire collected demographic data and assessed PA and resilience.

#### 2.3.1. Physical Activity

The International Physical Activity Questionnaire (IPAQ)-short form was used for the participants to self-report their engagement in different intensity levels of PA in the past seven days [23]. The IPAQ-short form consists of seven items asking questions about days and minutes for engaging in vigorous PA, moderate PA, walking, and sedentary behavior over the last seven days. For example, questions include “During the last 7 days, on how many days did you do vigorous physical activities like heavy lifting, digging, aerobics, or fast bicycling?” with the follow up “How much time did you usually spend doing vigorous physical activities on one of those days?”. Days per week, hours and minutes were recorded answers to these questions. The same format was used for moderate PA and walking. The last question pertained to sitting and asked participants “During the last 7 days, how much time did you spend sitting on a week day?”. The data were reported using the continuous variable MET-min/week, which was calculated by multiplying an intensity factor by the minutes and days reported. For walking, the intensity factor is 3.3, moderate PA is 4.0 and vigorous PA is 8.0. The total PA (TPA) MET-min/week was calculated by summing the scores of walking, moderate PA (MPA), and vigorous PA (VPA) MET-min/week. The Cronbach alpha was 0.75, 0.76 and 0.84 at pre-test, post-test, and follow up, respectively.

#### 2.3.2. Resilience

The Connor–Davidson Resilience Scale (CD-RISC-10) is a 10-item unidimensional scale that was used for the participants to self-rate their resilience [24,25]. This scale has been used in past studies examining physical activity and resilience [7,26,27]. The scale focuses on measuring the ability to recover from life’s obstacles. Example questions included “I am able to handle unpleasant or painful feelings like sadness, fear, and anger”, and “I tend to bounce back after illness, injury or other hardships”. Participants were asked to rate the degree to which the statements felt true to them and responded to each question using a 5-point Likert scale that ranges from not true at all (0) to true nearly all the time (4). Based on the implementation instructions, scores are summed, with higher scores indicating more resilience and lower scores indicating lower resilience [25]. Scores can possibly range from 40 (most resilient) to 0 (least resilient). The Cronbach alpha was 0.84, 0.90, and 0.86 at pre-test, post-test, and follow up, respectively.

### 2.4. Statistical Analysis

A total of 77 participants completed the intervention. Of the 77 participants, 57 participants completed the follow up questionnaire. Two participants were excluded due to missing outcome variable data. An independent-sample t-test revealed no significant differences at pre-test in demographic and outcome variables between the WeActive and WeMindful groups. Logistic transformations were performed on PA data due to non-normality. Skewness and kurtosis were checked for normality in resilience scores. Normality was assessed based on scores less than 1.96 [28]. A repeated-measures analysis of variance (ANOVA) was used to examine effects of intervention on physical activity and resilience by group. The between-subject factor was intervention group and the within-subject factor was time (pre-test, post-test, and follow up). All data analyses were performed using IBM SPSS 27.0 (IBM Corp, Armonk, NY, USA), and the significance level was set at *p* < 0.05.

## 3. Results

### 3.1. Intervention Effects on Physical Activity

Table 2 illustrates the descriptive statistics of TPA, VPA, MPA, and walking at pre-test, post-test, and follow up by group. Table 3 presents the results of the repeated-measures ANOVA for all PA variables. As seen in Table 3, for TPA, there was a significant main effect of time (*F* = 8.20, *p* = 0.001, *η*^2^ = 0.081). Although the WeActive group showed an increase in TPA from pre-test to post-test and from post-test to follow up (see Figure 1a), no significant increase between the two time points were found (see Table 4). On the other hand, a post hoc test revealed a significant difference between pre-test and follow up (*t* = 2.6, *p* = 0.002) and post-test and follow up (*t* = 3.6, *p* = 0.033) for the WeMindful group (see Table 5). In other words, the WeMindful group significantly decreased their total physical activity at follow-up test compared to both pre-test and post-test (see Figure 1a). Further, the results of repeated-measures ANOVA revealed no significant main effect of group or interaction of time with group for TPA (see Table 3).

Regarding VPA, there was a significant main effect of time (*F* = 3.90, *p* = 0.027, *η*^2^ = 0.025) (see Table 3). A post hoc test indicated that the WeActive group showed a significant increase in VPA from pre-test to post-test (*t* = −2.5, *p* = 0.037) (see Table 4). During the intervention, the WeActive group increased their VPA significantly. While not significant, the WeActive group continued to increase their VPA at follow up compared to pre-test (Figure 1b). There was no significant main effect of group or interaction of time with group for VPA (see Table 3). The WeMindful group showed no significant changes in VPA (see Table 5), but showed slightly increased VPA from pre-test to follow up (see Figure 1b)

For MPA, there was a significant interaction effect of time with group (*F* = 4.81, *p* = 0.01, *η*^2^ = 0.038) (see Table 3). A post hoc test showed that the WeActive group significantly increased MPA over time from pre-test to follow-up test as compared to the WeMindful group (*t* = −2.6, *p* = 0.033) (see Table 4). In other words, the WeActive group exhibited a significant increase in MPA from pre-test to follow up. However, the WeMindful group showed a decrease in MPA (see Figure 1c), although this was not significant (see Table 5). There was no significant main effect of group or time for MPA (see Table 3).

Finally, for walking PA, there was a significant main effect of time (*F* = 8.67, *p* < 0.001, *η*^2^ = 0.083) and interaction of time and group (*F* = 3.32, *p* = 0.047, *η*^2^ = 0.034) (see Table 3). A post hoc test revealed that the WeMindful group showed a significant decrease in walking PA from pre-test to follow-up test and post-test to follow-up test (see Table 5). In other words, the WeMindful group initially increased their walking PA during the intervention and then decreased from post-test to follow up (see Figure 1d). The WeActive group in contrast did not show a significant change in walking PA (see Figure 1d and Table 4).

### 3.2. Intervention Effects on Resilience

Table 6 illustrates the descriptive statistics of resilience at pre-test, post-test, and follow up by group. Table 7 presents the results of the repeated-measures ANOVA for resilience.

There was no significant main effect of the group or interaction effect of time with group (see Table 7). However, there was a significant main effect of time (*F* = 4.15, *p* = 0.024, *η*^2^ = 0.016) (see Table 7). Post hoc analysis revealed that both the WeActive and WeMindful groups showed a significant increase in resilience from pre-test to follow-up test (*t* = −2.735, *p* = 0.023) and (*t* = −2.535, *p* = 0.037), respectively (see Table 8). Although there was an increase (see Figure 2), there was no significant change from pre-test to post-test and post-test to follow up for both groups. In comparing the WeActive and WeMindful groups to normative data of undergraduate students, mean scores for both groups were within one SD of average at each time point [24].

## 4. Discussion

This study aimed to examine the immediate and short-term effects of virtual aerobic-strength exercise and mindful exercise interventions on physical activity intensity engagement and resilience among college students. As partly hypothesized, the WeActive group showed an increase in TPA at each time point although not reaching a significant level; however, the WeMindful group saw an initial increase and then a decrease in TPA. In relation to intensity, the WeActive group increased VPA significantly from pre-test to post-test and MPA from pre-test to follow up. The WeMindful group did not show any significant changes in VPA and MPA at any time point. However, the WeMindful group saw an initial non-significant increase in walking PA from pre-test to post-test followed by a significant decrease from post-test to follow up. The WeActive group did not show any significant changes in walking PA at any time point.

Congruent with the hypothesis, both groups significantly increased resilience from pre-test to follow up. This increased resilience demonstrated in both the WeActive and WeMindful groups may be due to a biological mechanism associated with exercise. Individuals who are more physically active tend to have lower cortisol levels, which are associated with lower levels of depression [29]. Additionally, it was found that after individuals were instructed to refrain from exercise, lower parasympathetic nervous system activity was observed along with reported negative mood [29]. Those who exercise may have a lower pre-test stress and can therefore handle increases in stress better than inactive individuals. Additionally, increased aerobic fitness is associated with increased prefrontal gray matter [29]. This increase could allow subjects to plan better and think more clearly when confronted with an obstacle. Lending support for the present results, previous studies of over 700 University students in Spain have found that higher levels of moderate PA are associated with greater resilience [30,31]. Further, a cross-sectional study in Spain during a national COVID-19 lockdown showed that older adults who engaged in MVPA scored better on the resilience scale [15].

Additionally, a study found that VPA is associated with increased resilience; however, there was no such relationship for MPA [13]. This indicates that VPA may account for a larger contribution to the increased resilience found in the WeActive group. Further studies on the specific adaptations from vigorous versus moderate physical activity should be examined to better understand how different PA intensities influence resilience.

The WeActive group saw a significant increase in MPA from pre-test to follow up. The WeActive intervention sessions included MPA exercises. A previous randomized controlled trial of 110 sedentary adults found that a web-based PA intervention including self-monitoring, pedometers and social elements delivered through social media significantly increased moderate–vigorous PA as a primary outcome; however, there were no significant changes in VPA [32]. This increase was not sustained at the 20 week follow up [32]. The participants in WeActive may have enjoyed an instructor-led class instead of self-monitoring. Moreover, 145 undergraduates retrospectively reported significantly less VPA in college than in high school [33]. Combined with the findings of the current study, university students may engage in more and prefer MPA over VPA. The discrepancy between WeActive’s significant increase in MPA while WeMindful did not see such an increase may be due to the higher intensity of the WeActive PA intervention. While the participants had access to both intervention materials after the intervention period was over, the WeActive participants completed the strength and aerobic, moderate-intensity, circuits while the WeMindful participants completed lower-intensity PA. Therefore, the WeActive participants who are familiar with the MPA materials are more likely to continue to engage in MPA after the intervention than the WeMindful participants.

The WeMindful group engaged in an 8 week virtual yoga exercise and mindfulness meditation. A significant increase in resilience was found between pre-test and follow up. Consistent with the present results, a short-term increase in resilience was found among forty-four participants who engaged in six weekly mind-body intervention sessions [34]. Similarly, a study found that after engaging in a 9 week mind-body intervention, 279 students showed a significant increase in resilience and mindfulness, compared to 247 students who did not engage in the intervention [35]. Mindfulness was sustained at one-year follow up, suggesting positive long-term impacts of the intervention [35]. Contrary to the current findings, an intervention study with 232 medical students during one semester in New Zealand did not find a statistically significant difference between a weekly peer-led mindfulness group and the control group [36]. One possible reason for the contradiction could be the social component. The peer leaders in the study by Moir et al. were chosen through applications and nominations. A positive, respectful, and supportive group atmosphere is crucial to mind-body interventions [35]. It is possible that the group dynamics were less supportive as it is described that there may have been compensatory rivalry between the intervention and non-intervention groups [36]. More research should be conducted into group social dynamics and their impact on mindfulness training.

It is worth noting that one advantage of the WeActive and WeMindful intervention programs was their virtual delivery. Participants in Lowenthal et al. expressed that virtual delivery would have increased conveniency for busy individuals, possibly by cutting down on travel time [34]. The participants in the WeActive and WeMindful groups had the added benefit of the second weekly session to be asynchronous. This means that students could complete the second weekly exercise session at the most convenient time for their schedules. Further, in a previous virtual aerobic and yoga intervention study of 71 university students, participants reported that the intervention was acceptable, appropriate, and feasible for them [16]. Virtual delivery likely contributed to this positive feasibility as it is a flexible option for busy participants who may have varying schedules week to week.

A strength of the current study is that it included students from various study years. There is a relatively even distribution of first, second, third, fourth, masters and doctoral students. Each year brings about new challenges and learning about resilience throughout the university journey is an advantage. It is advised that the findings from this study are used to inform university authorities to increase access to mindfulness resources and aerobic and resistance exercises to increase the resilience of their student population.

However, the virtual setting also has a limitation. The participants muted their sound and camera throughout the WeActive and WeMindful sessions. The instructors were unable to answer questions or check physical form. Although the virtual delivery allows for schedule flexibility, another limitation is the difficulty in assessing engagement accuracy. Participants were allowed to complete the second session asynchronously each week and self-report completion. This means the engagement rate could be subject to self-report bias. Along with engagement rate, PA and resilience were both measured through a self-report questionnaire which could be affected by self-report bias. Another limitation of this study was that it was conducted during the COVID-19 pandemic. Students were under a lot of unforeseen stress throughout the intervention and follow up. Some students may have been facing large novel obstacles that could have affected their perceived resilience. While the intervention being held during a pandemic allows for a snapshot in time, it can be less generalizable to a post-pandemic situation. The addition of a control group to this study would also add a better understanding of the effects of aerobic-resistance training and mindful yoga exercises had on resilience. Additionally, in the current study, the Cronbach alpha for the IPAQ ranged from 0.75 to 0.84. In previous research, the IPAQ short form had a reliability coefficient ranging from 0.71 to 0.89 for university students [37]. The current study falls within this range. This presents as a possible limitation as previous research has indicated that a reliability coefficient of 0.80 or greater is required to examine mean differences among groups [38]. Moreover, the lack of diversity in participants is a limitation of this study. Cisgender females made up 84% of the participants and 65% of the participants were white. According to a cross-sectional study of 77 undergraduate students, 79.2% who were white, it was found that a significantly higher proportion of male students were adequately physically active based upon accelerometer data [39]. The high percentage of cisgender females may have caused bias in the current study results. In a cross-sectional study of American university students, it was found that reported MPA was not significantly different between men and women; however, men engaged in significantly more VPA. [40]. Therefore, a large representation of cisgender females could cause bias; however, this study adds to the literature on exercise intervention effects on female students. Additionally, a cross-sectional study of 606 university students revealed that significantly more non-Hispanic white students met the MVPA recommendations by the 2018 Physical Activity Guidelines by the Department of Health and Human Services [40]. Thus, more research should be conducted on the ways in which student’s race influences exercise intervention effects.

## 5. Conclusions

The WeActive and WeMindful interventions significantly increased resilience in university students during a stressful time point in history. Further, there is evidence that this increased resilience was sustained in the short term after the intervention was terminated. Additionally, the WeActive intervention containing aerobic-resistance training exercises significantly increased MPA in the short term. The results of this study should be used to inform university health programs to increase students’ physical and mental health.

## Figures and Tables

**Figure 1 ijerph-19-04144-f001:**
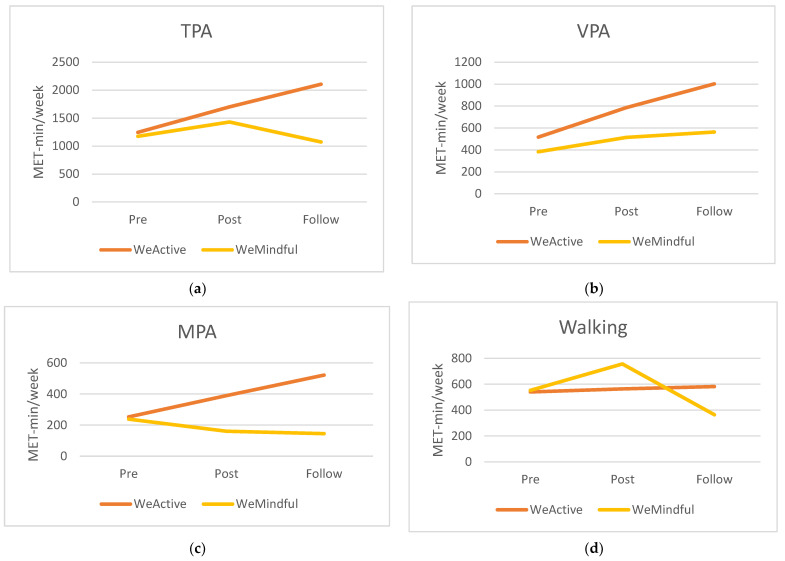
Descriptive statistics for both groups: (**a**) TPA; (**b**) VPA; (**c**) MPA; (**d**) walking.

**Figure 2 ijerph-19-04144-f002:**
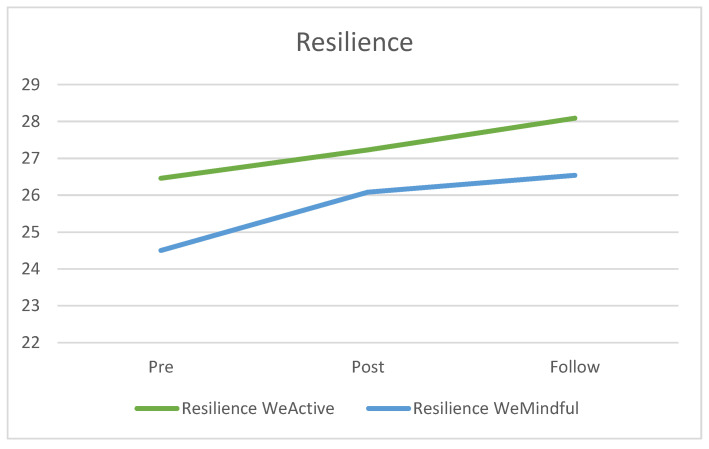
Average resilience scores for both groups at pre-test, post-test and follow up.

**Table 1 ijerph-19-04144-t001:** Demographic data of total participant group.

Variable		*n*	%
Gender	Cisgender female	46	84%
	Cisgender male	5	9%
	Transgender and gender non-conforming (TGNC)	4	7%
Race	Asian	11	20%
	Black or African American	3	5%
	White	36	65%
	Multiracial	5	9%
Ethnicity	Hispanic	3	5%
	Not Hispanic	52	95%
Education status	1st year	5	9%
	2nd year	6	11%
	3rd year	13	24%
	4th year	9	16%
	Master’s	8	15%
	Professional	0	0%
	Doctoral	14	25%
Group	WeActive	31	56%
	WeMindful	24	44%

**Table 2 ijerph-19-04144-t002:** Descriptive statistics of PA by group at pre-test, post-test, and follow up.

		WeActive		WeMindful	
Variable	Time Point	Mean	SD	Mean	SD
TPA	Pre-test	1244.79	1276.52	1174.08	1030.96
	Post-test	1700.21	1627.75	1430.21	884
	Follow up	2105.97	2253.91	1072.71	1382.84
VPA	Pre-test	516.57	772.31	383	612.07
	Post-test	785.45	937.59	513.33	558.34
	Follow up	1002.22	1273.52	563.33	892.63
MPA	Pre-test	252.78	398.09	328.22	320.7
	Post-test	389.41	819.42	160	154.69
	Follow up	521.67	558	145	318.65
Walk	Pre-test	540.38	503.96	552.75	839.54
	Post-test	563.91	466.08	756.94	618.18
	Follow up	582.08	633.27	364.38	534.57

**Table 3 ijerph-19-04144-t003:** Results of repeated-measures ANOVA for all PA variables.

Effects	Total			Vigorous			Moderate			Walking		
	*F*	*p*	*η^2^*	*F*	*p*	*η^2^*	*F*	*p*	*η^2^*	*F*	*p*	*η^2^*
Group	0.97	0.33	0.007	1.99	0.164	0.023	1.39	0.244	0.013	0.11	0.743	<0.001
Time	8.20	0.001 **	0.081	3.90	0.027 *	0.025	0.82	0.444	0.007	8.67	0.001 **	83
Time × Group	2.36	0.11	0.025	0.51	0.585	0.003	4.81	0.01 **	0.038	3.32	0.047 *	0.034

Note: * = *p* < 0.05; ** = *p* < 0.01.

**Table 4 ijerph-19-04144-t004:** Results of post hoc tests for all PA variables for the WeActive group.

Effects	TPA		VPA		MPA		Walk	
	*t*	*p*	*t*	*p*	*t*	*p*	*t*	*p*
Pre:Post	−1.899	0.149	2.532	0.037 *	−1.405	0.345	0.795	0.708
Post:Follow Up	1.755	0.195	0.301	0.951	−1.022	0.567	1.191	0.464
Pre:Follow Up	0.322	0.944	2.233	0.075	−2.585	0.033 *	0.596	0.823

Note: * = *p* < 0.05.

**Table 5 ijerph-19-04144-t005:** Results of post hoc tests for all PA variables for the WeMindful group.

Effects	TPA		VPA		MPA		Walk	
	*t*	*p*	*t*	*p*	*t*	*p*	*t*	*p*
Pre:Post	0.927	0.626	−1.504	0.297	−0.355	0.933	−1.304	0.399
Post:Follow Up	3.6	0.033 *	0.528	0.858	2.104	0.098	3.905	0.001 **
Pre:Follow Up	2.581	0.002 **	−0.75	0.735	1.85	0.163	2.887	0.015 *

Note: * = *p* < 0.05; ** = *p* < 0.01.

**Table 6 ijerph-19-04144-t006:** Descriptive statistics of resilience by group at pre-test, post-test, and follow up.

		WeActive		WeMindful	
Variable	Time Point	Mean	SD	Mean	SD
Resilience	Pre-test	26.46	6.14	24.5	5.91
	Post-test	27.23	8.18	26.08	5.22
	Follow up	28.09	6.08	26.54	4.2

**Table 7 ijerph-19-04144-t007:** Results of repeated-measures ANOVA for resilience.

Effects	Resilience		
	*F*	*p*	*η^2^*
Group	0.93	0.340	0.013
Time	4.15	0.024 *	0.016
Time × Group	0.16	0.822	0.001

Note: * = *p* < 0.05.

**Table 8 ijerph-19-04144-t008:** Results of post hoc for resilience by group.

Effects	WeActive		WeMindful	
	*t*	*p*	*t*	*p*
Pre:Post	−0.856	0.670	−1.362	0.368
Post:Follow Up	−1.069	0.537	−0.405	0.914
Pre:Follow Up	−2.735	0.023 *	−2.535	0.037 *

Note: * = *p* < 0.05

## Data Availability

The data that support the findings of this study are available on request from the corresponding author (W.C.). The data are not publicly available because they contain information that could compromise the privacy of research participants.

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
