# Peer review of "Enhancing Physical Activity and Psychological Well-Being in College Students during COVID-19 through WeActive and WeMindful Interventions"

_ijerph, 2022, doi:10.3390/ijerph19074144_

Round 1

Reviewer 1 Report

In my viewpoint, the manuscript tackled an interesting topic for the field of physical activity promotion by considering technology for interventions. However, there are a series of concerns and questions that need attention.

Comment 1. Please, provide a reference (i.e., systematic review or meta-analysis study) to strongly justify this sentence: “Physical activity is crucial in developing and strengthening resilience.”

Comment 2. Please, provide a solid reference such as systematic review or meta-analysis study to support this sentence: Despite PA’s beneficial effect on resilience, university students tend not to be 51 engaging in the recommended amounts of exercise.

Comment 3. It is missed a stronger rationale for the implementation of virtual-based PE intervention programmes.

Comment 4. Regarding the hypotheses for this research, the authors should be more cautious given that the absence of a (waitlist) control group. In this same vein, there is a need to justify why a control group was not considered.

Comment 5. Please, provide a rationale for the unbalanced number for the two intervention groups. Moreover, the little number of cisgender male compared to cisgender female, and transgender and TGNC people could represent a limitation in interpreting the results obtained.

Comment 6. Please, provide more detailed information on the instructor(s) who conducted the two intervention programmes. For instance: education and qualification, previous experience in (virtual) PA programmes.

Comment 7. Concerning intervention, it is required to detail if there was some type of external assessment by external observer to ensure the correct implementation of both programmes.

Comment 8. Reliability for IPAG represents a possible limitation.  Previous research indicates that values equal to 0.80 or higher are required when the purpose is to examine mean differences among groups (Viladrich et al., 2017).

Comment 9. Please, it is required to provide more information on the Connor-Davidson Resilience Scale. To illustrate this: a) is the instrument well-validated to PA settings? b) stem that precedes the items; c) a unidimensional or multidimensional measure; d) number of items comprising the instrument; and e) rationale for a global score instead of a mean score for the resilience variable.

Comment 10. Please, provide evidence in support of the normality assumption for resilience. Particularly, (Field, 2017) proposes that standardised values as high as 1.96 for skewness and kurtosis would gather evidence underpinning a normal data distribution.

Reference

Field, A. (2017). Discovering statistics using IBM SPSS statistics (5th ed.). SAGE Publications.

Viladrich, C., Angulo-Brunet, A., & Doval, E. (2017). A journey around alpha and omega to estimate internal consistency reliability. Anales de Psicologia/ Annals of Psychology, 33(3), 755–782. https://doi.org/10.6018/analesps.33.3.268401

Reviewer 2 Report

See attachment. 

Round 2

Reviewer 1 Report

I would like to thank the authors for the great labour in responding to every comment proposed. I feel fully satisfied with the current version of the manuscript.

Reviewer 2 Report

The reviewer thanks the authors for the opportunity to read and review this work. After revision, I believe the authors well addressed all my concerns in this manuscript. Well done.